# Factors and Methods for the Detection of Gene Expression Regulation

**DOI:** 10.3390/biom13020304

**Published:** 2023-02-06

**Authors:** Mengyuan Wang, Qian Li, Lingbo Liu

**Affiliations:** Institute of Hematology, Union Hospital, Tongji Medical College, Huazhong University of Science and Technology, Wuhan 430022, China

**Keywords:** gene expression regulation, detection methods, transcription factors, epigenetics

## Abstract

Gene-expression regulation involves multiple processes and a range of regulatory factors. In this review, we describe the key factors that regulate gene expression, including transcription factors (TFs), chromatin accessibility, histone modifications, DNA methylation, and RNA modifications. In addition, we also describe methods that can be used to detect these regulatory factors.

## 1. Introduction

Gene expression is the conversion of the genetic information carried by a gene into a biologically functional protein through key processes such as DNA transcription and RNA translation. The functional products of some genes are RNAs and do not need to undergo RNA translation; however, most genes need to undergo translation to produce proteins in order to perform their biological functions. The regulation of gene expression occurs at multiple levels: transcriptional, post-transcriptional, translational, and post-translational. Transcription is a fundamental aspect of gene expression and involves transcription factors (TFs), an important class of proteins that can recognize and bind to specific DNA sequences in the promoter region of genes. Conrad Waddington was the first to introduce the term “epigenetics” [1], a process that refers to changes in the expression level and function of genes that induce changes in heritable traits in the absence of changes in the DNA sequence.

Defining the binding sites of TFs, predicting potential TFs, and mapping epigenetics are crucial for understanding gene expression. With the development of second- and third-generation sequencing technologies, the methods for detecting TFs, chromatin accessibility, histone modifications, DNA methylation, and RNA modifications are becoming increasingly sophisticated. Here, we describe the effects of TFs and epigenetic modifications, including chromatin accessibility, histone modifications, DNA methylation, and RNA modifications, on gene expression, with particular focus on transcriptional regulation and the relationships of these factors with certain biological processes and diseases. We also describe the key methods used to detect these factors.

## 2. TFs

The interaction between TFs and DNA is the basis of transcriptional regulation, in which the binding sites of TFs on DNA are essential for the functionality of TFs. Therefore, the identification of TF binding sites (TFBS) is indispensable if we are to fully understand transcriptional regulation [2].

### 2.1. Genome-Wide Methods for Identifying TFBS

#### 2.1.1. Chromatin Immunoprecipitation

Chromatin immunoprecipitation (ChIP) is a technique that uses antibodies to immunoprecipitate target proteins, thereby enriching and identifying the gene sequence to which the target protein is bound [3]. First, formaldehyde is used to crosslink protein with DNA; then, chromatin fragmentation is performed by sonication [4] or micrococcal nuclease (MNase) digestion [5]. This is then followed by immunoprecipitation of protein–DNA complexes using antibodies to the target protein. Finally, the crosslinking is removed, and the DNA fragments are analyzed. ChIP-seq (chromatin immunoprecipitation followed by sequencing) combines ChIP and next-generation sequencing, thus allowing the identification and quantification of genetic sequences bound by a target protein at the base-pair level [6] (Figure 1a). In 2007, Johnson et al. [7] used ChIP-seq to discover genome-wide binding sites for the transcription factor neuron-restrictive silencer factor (NRSF) to DNA. ChIP-seq is applicable not only to TFs but also to other DNA-binding proteins, and can capture binding information relating to various histone modifications on the genome. However, ChIP-seq has several limitations. For example, ChIP-seq is dependent on antibody quality and requires ChIP-grade antibodies, which are sometimes not commercially available. Furthermore, this technique requires millions of cells, is time-consuming and labor-intensive, and also lacks robustness due to experimental variation [6]. Dirks et al. [8] invented the plug-and-play (PnP) ChIP-seq microfluidic platform, which can perform robust and standardized ChIP-seq on hundreds to thousands of cells in only 1 day.

Researchers have also developed an efficient single-cell (sc) ChIP-seq technique (sc-ChIP-seq); Wang et al. [9] developed a universal, high-quality, high-throughput, and easy-to-use sc-ChIP-seq method by combining barcoding and targeted chromatin-release techniques (CoBATCH), which was proven to be very useful for studying cellular heterogeneity, although 10,000 cells are required to perform this technique. In another study, Ai et al. [10] developed another single-cell ChIP technique that can be used to capture starting volumes as low as 100 cell samples. This technique is referred to as single-cell simultaneous indexing and tagmentation-based ChIP-seq (sc-itChIP-seq) and does not rely on specialized equipment; however, this technique has a higher mapping rate and a greater sensitivity.

#### 2.1.2. Cleavage under Targets and Release Using Nuclease and Cleavage under Targets and Tagmentation

Over recent years, other techniques have been developed: Cleavage Under Targets and Release Using Nuclease (CUT&RUN) and Cleavage Under Targets and Tagmentation (CUT&Tag); these have become very popular. CUT&RUN is based on chromatin immunocleavage (ChIC) technology, which uses the pA-MN fusion protein, which consists of protein A and MNase to cleave the DNA region bound to a target protein [11]. In CUT&RUN, cells are first immobilized on magnetic beads; then, they are incubated sequentially with an antibody and pA-MN (which can bind to the antibody). At 0 °C, Ca2+ is added and MNase cleaves DNA on both sides of the binding site. Subsequently, DNA is released and diffuses into the supernatant from which it is extracted and sequenced [12,13] (Figure 1b). Compared with ChIP-seq, CUT&RUN has several advantages. First, this technique is performed in situ without cross-linking; this can avoid the generation of false-positive sites formed by the excessive cross-linking of formaldehyde [14,15]. Second, this technique is easy to perform, can be completed in 1 day, and is cost-effective. Third, this technique has a low background and a high signal-to-noise ratio. Fourth, CUT&RUN requires a small number of cells and used 100 cells and 1000 cells to map genome-wide DNA binding sites for histone modifications H3K27me3 and TF CTCF, respectively [13]. However, some samples do not contain a sufficient number of cells for this technique; consequently, ultra-low input CUT&RUN (uliCUT&RUN) was developed by optimizing the CUT&RUN technique. This technique can be applied to a much smaller number of cells. Hainer et al. [16] used uliCUT&RUN to acquire the binding profiles of several DNA binding proteins, including TFs, histone modifications, and chromatin-modifying enzymes from 50 cells. In addition, the authors generated the first genome-wide binding profiles of the transcription factors CTCF, SOX2, and NANOG, in single cells.

The DNA obtained from CUT&RUN requires adapter ligation prior to sequencing, thus increasing the workload required. As a result, Kaya-Okur et al. [17] developed the CUT&Tag technique from CUT&RUN and replaced MNase with hyperactive Tn5 transposase loaded with sequencing adapters [18]. Tn5 cleaves DNA on both sides of the target protein while ligating sequencing adapters; this strategy can speed up the subsequent library-building step. In addition, CUT&Tag is suitable for single-cell platforms [17,19] (Figure 1c). Droplet-based single-cell CUT&Tag (scCUT&Tag) has been used to generate binding maps of histone modifications and two transcription factors (OLIG2 and RAD21) at the single-cell level [20,21].

#### 2.1.3. DNA Affinity Purification Sequencing

DNA affinity purification sequencing (DAP-seq) was invented by O’Malley et al. [22]. In this technique, genomic DNA is first fragmented and then ligated to sequencing adaptors. Next, TFs fused to Tag are expressed in vitro and bound to magnetic ligand-coupled beads, which prevents the Tag-fused TFs from being eluted, while non-specific proteins are washed away. Then, genomic DNA and transcription factor fusion proteins are incubated, unbound DNA fragments are washed away, TF–DNA complexes are purified using magnetic separation of the affinity tag, and finally the transcription factor-bound DNA fragments are enriched for sequencing (Figure 1d). Although this technique has only been available for a short time, studies have been conducted to detect TFBS in both fungi [23] and flowers [24]. DAP-seq does not require specific antibodies, is relatively low-cost, and can be used for large-scale transcription factor studies. However, DAP-seq has some technical limitations, including a low success rate (~30%) and the inability to detect the effects of histone modifications and chromatin accessibility [25].

### 2.2. Detection Methods for TFs

The experimental techniques described in the previous section all required a selected TF; however, sometimes we need to target TFs that play a role in the transcriptional regulation of specific genes. This problem can be solved by DNA pull-down; in this technique, nuclear proteins are captured by the promoter fragment bound to the magnetic beads. Then, mass spectrometry identifies and screens for proteins that bind specifically to the promoter fragment [26]. In addition, gene promoters can be analyzed by applying the UCSC Genome Browser database [27] and the JASPAR database [28] to predict potential TF and TFBS. Once potential TFs have been identified, ChIP is used to analyze whether the TF binds directly to the gene; dual-luciferase reporter assays are then used to verify the transcriptional activity of the target TF [29,30].

## 3. Chromatin Accessibility

Approximately 147 bp of DNA wraps around a core histone octamer in a left-handed super-helical manner to form a nucleosome core particle. These nucleosome core particles are arranged like beads on a string by linker DNA and histone H1 links which bend and fold further to form chromatin [31,32]. Chromatin is a dynamic structure that influences chromatin accessibility during gene transcription, DNA replication, and DNA damage repair, thus exerting a regulatory effect on these processes [33]. For example, during DNA transcription, histone octamers spontaneously unwrap, thus exposing the TFBS covered by nucleosomes; the TFBS bind to TFs to promote gene transcription [34].

Chromatin accessibility refers to the extent to which macromolecules within the nucleus can physically make contact with chromatin DNA, and is primarily related to the organization and occupancy of nucleosomes and DNA-binding molecules, including TFs, RNA polymerases, linker and core histone proteins, as well as insulator proteins [35]. The study of chromatin accessibility stems from the discovery of chromatin-specific sites with periodic hypersensitivity to DNase I [36]. These sites are referred to as DNase hypersensitivity sites (DHSs). In these specific regions of the genome, chromatin structures remain open. Subsequent research found that functional cis-acting elements, such as enhancers, promoters, silencers, and insulators, are coupled to DNase I hypersensitive sites [37,38]. In the 1980s, it was found that when a gene is transcriptionally active, the chromatin region in which a gene is located is much more sensitive to DNase I degradation than other regions that are not transcriptionally active, thus suggesting that the sensitivity of chromatin to DNase I is related to gene transcription [39]. Accessible chromatin genomes represent 2–3% of the total DNA sequence and more than 90% of accessible chromatin regions are bound to TFs [37]. Accessible chromatin regions can be determined by their sensitivity to enzymatic methylation or cleavage [35].

### 3.1. Detection Methods for Chromatin Accessibility

#### 3.1.1. DNase I Hypersensitive Site Sequencing Subsubsection

DNase I hypersensitive site sequencing (DNase-seq) specifically cleaves accessible chromatin regions characterized by DHSs with optimal concentrations of DNase I; the resulting DNA fragments are then subjected to high-throughput sequencing [40] (Figure 2a). This method has the advantage of excellent specificity, throughput, and sensitivity, but also requires millions of cells and involves a complex sample preparation process and tedious enzyme titration steps to determine the optimal enzyme concentration [41]. Single-cell DNase-seq sequencing (scDNase-seq) has been developed to measure chromatin accessibility at the single-cell level [42,43].

#### 3.1.2. Micrococcal Nuclease Sequencing

In micrococcal nuclease sequencing (MNase-seq), MNase first acts as a nucleic acid endonuclease to cleave inter-nucleosomal DNA and then as an exonuclease to degrade the naked accessible DNA; the resulting DNA fragments are then sequenced (Figure 2b). This type of sequencing yields genomic regions occupied by nucleosomes and other regulatory factors, thus indirectly detecting chromatin accessibility and mapping nucleosomes; these are the major differences between MNase-seq and other methods for detecting chromatin accessibility [35,44]. There are several advantages to this method, including the standardization of digestion and data analysis and the ability to combine this technique with ChIP-seq to study nucleosome-binding regulators [45,46]. However, this technique also has some disadvantages: (1) it requires a large number of cells, (2) it necessitates time-consuming enzyme titration procedures to determine the optimal enzyme concentration [41], and (3) MNase prefers to cut AT-enriched regions [47,48]. Single-cell MNase-seq (scMNase-seq) was subsequently invented in 2018 [49].

#### 3.1.3. Sequencing-Based Assays for the Detection of Transposase-Accessible Chromatin

Assays for transposase-accessible chromatin involve sequencing (ATAC-seq) and utilize Tn5 transposase to insert into the accessible chromatin region; end-repairs are performed, and sequencing adaptors are added while cutting DNA fragments. Then, the DNA between the adaptors is amplified by polymerase chain reaction (PCR) and sequenced [50] (Figure 2c). ATAC-seq has numerous advantages: (1) only 500–50,000 cells are required; (2) this method is simple and fast; (3) this technique provides a large amount of data, including regulatory molecule binding regions, nucleosome positions, and genome-wide accessible chromatin regions; (4) ATAC-seq is expected to be used clinically to generate “personal epigenomic” profiles due to being rapid, informative, and requiring only a small number of cells, and (5) this technique can be combined with RNA-seq, ChIP-seq, and other techniques for multiomic analysis [35,41,50,51]. However, this technique also has a disadvantage: the number of cells determines the quality of sequencing. Too many or an insufficient number of cells leads to under-transposition or over-transposition [52]. In 2015, Cusanovich [53] and Buenrostro [54] successively proposed the detection of chromatin accessibility by single-cell ATAC-seq (scATAC-seq) and revealed the heterogeneity of chromatin accessibility between cells. Subsequently, scATAC-seq was gradually developed; this was followed by several scATAC-seq methods [55,56,57,58,59].

#### 3.1.4. Nucleosome Occupancy and Methylome Sequencing

Unlike the three methods described above, which all use enzymatic cleavage, nucleosome occupancy and methylome sequencing (NOMe-seq) uses methylation modification. NOMe-seq uses GpC methyltransferase (M.CviPI) to methylate GpC dinucleotides in accessible chromatin to GpCm without endogenous background. Bisulfite treatment can distinguish GpC from GpCm, and after sequencing, information relating to accessible chromatin regions can be obtained (Figure 2d). In addition, sulfite treatment can distinguish CpG from CmpG and endogenous DNA methylation information can be obtained from the same DNA strand after sequencing. The advantage of this method is that it does not involve enrichment bias and can generate data relating to both chromatin accessibility and DNA methylation; however, cellular requirements are high [35,60,61,62].

## 4. Histone Modifications

Histone modifications are covalent modifications of histone N-termina “tails” which project from the nucleosome [63]. In addition to common acetylation, methylation, phosphorylation, ubiquitination, and sumoylation, there are several other histone modifications, including ADP-ribosylation, deimination, proline isomerization, propionylation, butyrylation, crotonylation, 2-hydroxyisobutyrylation, malonylation, succinylation, formylation, citrullination, hydroxylation, O-GlcNAcylation, and lactylation [64,65]. In the 1960s, Allfrey et al. [66] linked histone acetylation and methylation to the regulation of gene expression. Histone modifications were subsequently found to play an important role in the direct control of gene expression [67]. Modifications in the core regions of histones are being increasingly studied. Research has found that modifications of histone cores can directly affect transcription; for example, H4K16ac is known to activate transcription, while H3K27me3 silences transcription [68,69].

### 4.1. Acetylation

Histone acetylation is almost always associated with transcriptional activation. Histone acetylation is modified by transferring the acetyl group from acetyl-coenzyme A (acetyl-CoA) to the ε-amino group of the N-terminal lysine side chain of histones via histone acetyltransferase (HAT). There are three members of the HAT family: the GNAT family, the MYST family, and the CBP/p300 family [70]. Deacetylation is catalyzed by histone deacetylase (HDAC), which is divided into four classes: Class I HDAC (HDAC 1, 2, 3, and 8), Class II HDAC (HDAC 4–7, 9–10), Class III HDAC (SIRT 1–7), and Class IV (HDAC11) [71]. Histone acetylation regulates gene expression in two ways: (1) the addition of acetyl groups neutralizes the positive charge of the lysine amino group and opens the compact chromatin structure, thus facilitating TF binding to DNA and thus promoting gene transcription [63]; and (2) the recruitment of regulatory proteins containing bromodomain and YEATS-domain that recognize acetylated lysine to regulate gene transcription [63,72].

### 4.2. Methylation

Histone methylation can either activate or repress transcription [63]. Histone methylation is a methylation modification catalyzed by K-methyltransferase (KMT) and protein arginine methyltransferase (PRMT) for lysine and arginine residues, respectively. There are two types of PRMT: the first type catalyzes the formation of monomethylarginine and asymmetric dimethylarginine, while the second type catalyzes the formation of monomethylarginine and symmetric dimethylarginine [73]. There are two demethylase structural domains: the LSD1 domain and the JmjC domain [63]. Histone methylation is recognized by the royal family of proteins (chromodomain, malignant brain tumor (MBT), conserved Proline and Tryptophan (PWWP), and Tudor domains), plant homeodomain (PHD), zinc-finger, and WD-repeat modules [63,72,74]. Histone methylation modifications cannot change the charge and regulate gene expression by recruiting proteins that recognize methylation modifications. For example, BPTF recognizes H3K4me3 via a PHD structural domain and subsequently binds to SNF2L ATPase to activate the expression of the H0XC8 gene [75].

### 4.3. Phosphorylation

Histone phosphorylation refers to the transfer of a phosphate group from ATP to the amino acid hydroxyl group by protein kinase. The reverse reaction can be achieved by the hydrolysis of the phosphate group by protein phosphatase. Phosphorylation regulates gene expression in a similar way to acetylation, first by changing the compact structure of chromatin by neutralizing the positive charge of histones, which in turn promotes TF binding to DNA [76]. Second, phosphorylation modifications are recognized by a structural domain in the 14-3-3 protein [63].

### 4.4. Methods for Identifying Specific Histone Modification Binding Sites

ChIP-seq is widely used in biological studies of histone modifications, and can identify DNA sequences bound to specific histone modifications and determine their location in the genome. Histone–DNA complexes can be precipitated by antibodies specific for histone modifications; then, the DNA is sequenced by high-throughput techniques [67,77]. However, there are still obstacles associated with antibody-based ChIP that need to be overcome. For example, antibodies can cross-react with similar modifications on histones; furthermore, epitope occlusion remains a problem, as does cost, and it is proving difficult to produce and validate antibodies. In addition to ChIP-seq, other methods that can be used to detect TFBS include CUT&RUN [78] and CUT&Tag [79]; these methods can both detect the binding sites of specific histone modifications.

### 4.5. Methods for Identifying Histone Modifications

ChIP-seq, CUT&RUN, and CUT&Tag are based on histone modifications of interest to determine their binding sites in the genome [13,21,80]. Mass spectrometry (MS)-based proteomics is also a powerful tool for identifying histone modifications [81]. In this technique, histone modifications are identified by first isolating and purifying histones. These are then detected by a range of mass spectrometry-based strategies such as bottom-up, top-down, or m-down. In the bottom-up strategy, the target protein is isolated and then enzymatically digested into peptides containing multiple amino acid sequences. These peptides are then detected by mass spectrometry, and information relating to protein sequence and modification is obtained by conventional collision induced dissociation (CID). Bottom-up MS is widely used, but most bottom-up MS techniques analyze short classes of peptides and do not allow for the detection of long-distance histone modification combinations that are concurrently present on individual histone tails, separated by long distances [82]. To solve this problem, top-down and middle-down MS were developed. Electron capture dissociation (ECD) and electron transfer dissociation (ETD)-based top-down strategies introduce intact proteins into a mass spectrometer for the identification of combined histone modifications [83]. The middle-down strategy is also based on ECD/ETD, in which proteins are digested into long peptides by specific enzymes for the identification of long-distance combined histone modifications [84].

## 5. DNA Methylation

At least 17 DNA modifications have been reported; of these, the 5mC modification—also known as the fifth base modification—is one of the most abundant, common, and widely studied DNA modifications [85]. DNMT1 (DNA methyltransferase 1), DNMT3A (DNA methyltransferase 3A), and DNMT3B (DNA methyltransferase 3B) are canonical DNA methyltransferases that transfer the methyl group of S-adenosylmethionine to the fifth carbon atom of cytosine [86]. Of these, DNMT3A and DNMT3B mediate de novo DNA methylation while DNMT1 maintains DNA methylation during DNA replication. There are two types of DNA demethylation: passive demethylation and active demethylation. Passive demethylation means that in the absence of functional DNA methylation maintenance machinery, 5mC is passively diluted by DNA replication [87]. Ten–eleven translocation (TET), thymine DNA glycosylase (TDG), and base excision repair (BER) are all known to mediate active demethylation. TET mediates the progressive oxidation of 5mC to 5hmC, 5fC, and 5caC. TDG specifically excises 5fC and 5caC to form an abasic site; then, the abasic site is repaired by BER to restore an unmodified cytosine [87,88] (Figure 3).

5mC has been associated with a variety of biological functions. Methylation was traditionally thought to be associated with the repression of gene expression. DNA methylation can prevent TFs from binding to DNA and thus inhibits gene transcription [89]. Methyl-CpG-binding proteins (MBP) specifically bind to methylated CpG sites. The binding of MBP to methylated sequences can inhibit the binding of TFs to DNA on the one hand and recruit and interact with HDAC on the other hand, thus causing chromatin condensation [90]. It was subsequently found that the regulation of gene transcription by methylation was related to the location of methylation. Specifically, the methylation of promoters were found to repress gene transcription while the methylation of gene bodies was found to be positively associated with gene expression [91].

DNA methylation is known to change with aging. Alterations in DNA methylation have been detected in many age-related diseases [92], including coronary heart disease [93], type 2 diabetes [94], osteoarthritis [95], Alzheimer’s disease [96], and cataracts [97]. Alterations in DNA methylation have also been detected in the early stages of tumorigenesis [98]. Furthermore, DNA methylation profiles are specific to the origin of a tumor; thus, DNA methylation is considered a biomarker for early diagnosis [99].

### 5.1. Detection Methods for DNA Methylation

Koch et al. [100] detected approximately 1800 DNA methylation-based biomarkers of cancer by searching PubMed; however, less than 1% (14/1800) of these biomarkers have been translated into commercially available drugs and tested in clinical trials. One particular obstacle is determining the exact location of DNA methylation. It is vital that we can identify the exact location of DNA methylation in the genome. In the following section, we summarize the methods that can be used for localization analysis; these techniques mostly involve next-generation sequencing and third-generation sequencing.

#### 5.1.1. Bisulfite Sequencing and Its Derivatives

Bisulfite sequencing (BS-seq) is a localization method that combines the bisulfite conversion of DNA methylation modifications with next-generation sequencing to achieve single-base resolution measurements of the methylation status of a whole genome [101]. After treatment with bisulfite, cytosine (C), 5-formylcytosine (5fC), and 5-carboxylcytosine (5caC) are deaminated to uracil (U) and read as thymine (T) in subsequent sequencing analyses. However, 5-methylcytosine (5mC) and 5-hydroxymethylcytosine (5hmC) are both resistant to this chemical conversion and are read as cytosine (C) [102] (Figure 4). This method can be used to locate and analyze 5mC and 5hmC in the genome, but it is not possible to distinguish between 5mC and 5hmC. Due to this issue, researchers developed oxidative bisulfite sequencing (oxBS-seq) and TET-assisted bisulfite sequencing (TAB-seq). OxBS-seq can map and quantify 5mC at single-base resolution. In oxBS-seq, 5hmC is specifically oxidated to 5fC by potassium perruthenate (KRuO_4_) and is then deaminated to U by bisulfite and read as T, while 5mC is read as C [103] (Figure 4). In TAB-seq, glucose is introduced into 5hmC via β-glucosyl transferase (β-GT) to produce β-glucosyl-5-hydroxymethylcytosine (5gmC); 5gmC is not oxidized by TET and is sequenced as C, while 5mC is oxidized by an excess of TET to 5caC and is sequenced as T. Thus, 5mC sites can then be deduced by the subtraction of TAB from BS measurements [104] (Figure 4).

Reduced representation bisulfite sequencing (RRBS) is a method used to enrich CpG regions using methylation-insensitive restriction enzymes prior to bisulfite treatment. Restriction enzymes specific for CpG containing motifs ensure that the read sequence includes at least one informative CpG position [105]. Via the application of restriction enzymes, adapters, bisulfite treatment, PCR amplification and sequencing, CpG-rich regions are methylation-sequenced with single-nucleotide resolution and high throughput, thus yielding an efficient and cost-effective technique [106]. However, RRBS cannot target enhancers and CTCF sites that are located beyond CpG islands. Extended representation bisulfite sequencing (XRBS) was invented to enrich CpG dinucleotides in promoters, enhancers, and CTCF sites; this technique has significant advantages over RRBS in respect to efficiency, coverage, and sensitivity [107].

Although BS sequencing and its derivative methods are the gold standards for quantitative DNA methylation analysis, these techniques are associated with certain limitations. First, bisulfite treatment is a harsh chemical reaction that results in over 99% of DNA being lost due to DNA degradation. BS-induced DNA degradation leads to the depletion of unmethylated C-rich sequences; this produces uneven sequence coverage and directly results in an overestimation of the absolute values of 5mC [108]. Second, bisulfite sequencing depends on the complete conversion of unmodified cytosine to thymine. Unmodified cytosines make up approximately 95% of all cytosines in the human genome. Converting all of these positions to thymine significantly reduces sequence complexity, thus resulting in high overall bias, decreased mapping rates, uneven genome coverage, and increased sequencing costs [108,109]. Third, because bisulfite can convert single-stranded DNA but not double-stranded DNA, the conversion will be incomplete due to incomplete denaturation or reannealing [106,109].

#### 5.1.2. TET-Assisted Pyridine Borane Sequencing and Its Derivatives

The TET-assisted pyridine borane sequencing (TAPS) technique was first developed by Liu et al. [109]. The TAPS technique is a mild chemical reaction that allows for the direct detection of modified cytosines at the single-nucleotide level without affecting unmodified cytosines. In TAPS, 5mC and 5hmC are oxidized to 5caC by TET enzymes; then, 5caC is reduced to dihydrouracil (DHU) by pyridine borane. DHU is read as T in subsequent sequencing (Figure 5). However, TAPS is unable to distinguish 5mC from 5hmC and thus requires some improvements to be made. In TAPS with β-glucosyltransferase blocking (TAPSβ), the introduction of a glucose group into 5hmC causes glycosylation to 5gmC, thus protecting 5hmC from TET oxidation and pyridylborane reduction; eventually, 5hmC is read as C. During this process, 5mC is converted to 5caC and then to DHU and is finally read as T (Figure 5). In addition to the reduction of 5caC, diborane can transform 5fC to DHU via a reductive deformation/deamination mechanism; this mechanism formed the basis of chemical-assisted pyridine borane sequencing (CAPS). KRuO4, a reagent used in oxBS-seq, specifically oxidizes 5hmC to 5fC, which is then reduced to DHU by pyridine borane and read as T during sequencing; in contrast, 5mC remains unchanged and is read as C [109,110] (Figure 5).

TAPS is associated with some clear advantages. First, it involves a mild reaction and avoids the conversion of unmodified cytosines. Furthermore, compared to BS-seq, TAPS offers higher sequencing quality, more complete methylation information, and is associated with reduced costs [109]. Siejka-Zielińska et al. [111] optimized TAPS for cell-free DNA (cfDNA), a technique known as cfTAPS, to provide genome-wide cell-free methylome information with high quality and high depth using only 10 ng of cfDNA (1 to 3 mL of plasma). First, 10 ng of cfDNA is ligated to Illumina adapters and then mixed with 100 ng of vector DNA prior to TET oxidation and pyridine borane reduction. cfDNA is free-floating DNA that is found in blood plasma; this form of DNA is derived from cell death in various tissues of the body and consists of fragmented DNA molecules [112,113]. Circulating tumor-derived DNA has a methylation status similar to that of tumor tissue, thus enabling the development of cancer screening and localization tests [112]. cfTAPS makes early cancer detection clinically possible, and an important advantage of this method is the ability to determine information relating to tissue origin [111].

Based on TAPS, Cheng et al. [114] developed eeTAPS (endonuclease enrichment TAPS), which uses the Uracil-Specific Excision Reagent (USER) enzyme, an endonuclease that specifically cleaves DHU, to enrich methylated CpG sites (mCpGs) by cleaving TAPS-transformed DNA. The authors applied eeTAPS to model DNA and genomic DNA (gDNA) from mouse embryonic stem cells (mESCs) and found that TAPS and eeTAPS cover almost as many CpG sites (19.2 million and 20 million sites, respectively), thus demonstrating that eeTAPS can detect genome-wide methylation at single-CpG resolution. eeTAPS is also known to be cost-effective.

#### 5.1.3. Methylated DNA Immunoprecipitation

Two methods can be used for the immunoprecipitation and enrichment of methylated DNA: methylated DNA immunoprecipitation sequencing (MeDIP-seq) and methylated DNA binding domain sequencing (MBD-seq). MeDIP-seq uses a 5mC antibody to enrich methylated DNA fragments, which are then subjected to high-throughput sequencing to obtain a high-resolution whole-genome DNA methylation profile [115] (Figure 6). Shen et al. [116] subsequently improved the MeDIP-seq protocol to enable the methylome sequencing of cfDNA down to a concentration of 1–10 ng; this technique was named cfMeDIP-seq. This method enables minimal invasive examination for early tumor detection, cancer interception, and the classification of multiple types of cancer [117]. However, MeDIP-seq has some limitations to consider: (1) the lack of single-base resolution analysis and (2) the enrichment of only the methylated part of the genome [118]. A method that combines MeDIPseq and methylation-sensitive restriction enzyme sequencing (MRE-seq) was subsequently developed by Harris et al. [118,119] in which MeDIP-seq enriches and detects detect methylated CpGs while MRE-seq enriches and detects unmethylated CpGs.

As the methylated-CpG-binding domain (MBD) can specifically bind to methylated CpG dinucleotides, a number of assays have been developed; these depend on the MBD protein precipitated [120]. For example, Serre et al. [121] developed MBD-isolated genome sequencing (MiGS), which uses recombinant MBD of MBD2 protein to precipitate methylated DNA (Figure 6). Methylation DNA capture sequencing (MethylCap-seq) is based on the MBD of MeCP2 to capture methylated DNA [122] (Figure 6). The methyl-CpG immunoprecipitation (MCIp) assay utilizes a recombinant methyl-CpG binding and antibody-like fusion protein consisting of the MBD of human MBD2, a flexible linker polypeptide, and the Fc portion of human IgG1 [123] (Figure 6). DNA methylated-CpG island recovery assay sequencing (MIRA-seq) is based on the high affinity of the MBD2b/MBD3L1 protein complex for methylated CpG dinucleotides [124,125] (Figure 6). Among the currently known methyl-CpG binding proteins, MBD2b has the highest affinity for methyl-CpG. Furthermore, MBD3L1 can interact with MBD2b and enhance the binding of MBD2b to methyl-CpG, thus providing the MBD2B/MBD3L complex with a higher affinity for methyl-CpG [124].

#### 5.1.4. Nanopore Sequencing

Nanopore sequencing is a form of third-generation sequencing technology in which different electrical current signals are generated when the same nucleotide (with and without modification) passes through the nanopore. These different nucleotides passing through the nanopore will obstruct the original stable electrical current and generate a disordered electrical current. The difference in the degree of obstruction of the nanopore channel by different nucleotides causes a change in the characteristics of the corresponding electrical current, thus generating different current signals, so that these different nucleotides can be distinguished according to changes in electrical current [126]. Nanopore sequencing has been successfully used for the detection and mapping of C, 5mC, and 5hmC [127,128]. In addition, the nanopore device can also distinguish 5fC and 5caC, thus meaning that this technique can distinguish all five C5-cytosine variants [129,130]. Compared to other sequencing methods, nanopore sequencing offers the advantages of low-cost sample preparation, no requirements for polymerase and ligase, and producing long read lengths (>10 kb). Furthermore, nanopores can read genomic DNA directly without amplification, thus avoiding errors caused by amplification replication preferences [130,131].

#### 5.1.5. Single-Molecule, Real-Time (SMRT) Sequencing

SMRT sequencing is also a form of third-generation sequencing technology. In SMRT sequencing, DNA polymerase catalyzes four distinguishable and fluorescently labeled deoxynucleoside triphosphatases (dNTPs) to synthesize a nucleic acid strand that is complementary to the template; the DNA sequence is then determined by detecting the fluorescence bound to the dNTPs [132]. When unmodified bases and differently modified bases are incorporated on the template, the processing ability of DNA polymerase will be slowed down, resulting in an increase in the time interval between the current base and the next base incorporation. As unmodified bases and differently modified bases vary in terms of their incorporation kinetics, this method can be used for the detection of 5mC [133], 5hmC [134], N6-methyladenine (6mA) [135], and N4-methylcytosine (4mC) [135]. SMRT sequencing has the advantages of a long read length (an average read length of 13.5 kb) and high accuracy (99.8%) [136]. The use of SMRT for long cfDNA detection and methylation analysis in cancer patients has opened up a new method for long cfDNA-based cancer diagnosis [137].

## 6. RNA Modifications

At least 160 types of RNA modifications have been identified, including N6-methyladenosine (m6A), 2′-O-dimethyladenosine (m6Am), N1methyladenosine (m1A), 5-methylcytosine (m5C), 5-hydroxymethylcytosine (hm5C), N4-acetylcytidine (ac4C), pseudouridine (Ψ), and N7-methylguanosine (m7G) [138]. m6A is the most abundant internal modification in mRNA and long non-coding RNA. m6A is catalyzed by the methyltransferase complex (the m6A “writer”) [139,140,141,142], which consists of METTL3, METTL14, WTAP, VIRMA, HAKAI, ZC3H13, RBM15/15B, and METTL16 methyltransferase (the m6A “writer”) [143]. m6A is a reversible chemical modification that can be demethylated by demethylases (the m6A “erasers”), including FTO, ALKBH5, and ALKBH3 [144,145,146].

The function of m6A in the regulation of gene expression is multifaceted [147]. m6A regulates the alternative splicing of pre-mRNA [148] and the processing of 3′ untranslated regions (UTRs). Researchers found that the 3′ UTRs of hundreds of target mRNAs were elongated in HeLa cells featuring the knockdown of VIRMA and METTL3 [141]. m6A is also known to influence the cytosolic export of mRNA. The deletion of ALKBH5 accelerates nuclear RNA export, thus leading to a significant increase in cytoplasmic RNA levels in cells [145]. m6A can also stimulate and inhibit translation in a manner that is dependent on cell type and developmental state. m6A also enhances the expression of p21 at the translational level [149]; however, during the maturation of Xenopus oocytes, m6A can inhibit translation [150]. m6A can also promote mRNA degradation in mESCs [151] and regulate RNA–protein interactions to affect gene transcription [152].

m6A can affect a variety of biological processes. For example, m6A is required for viability in Arabidopsis and mice [153,154]. Although m6A is not essential for the viability of Drosophila, m6A-deficient flies have been shown to be less fertile [155]. m6A plays an important role in the differentiation of embryonic and adult stem cells [156]. m6A and its associated regulators also play important roles in various human cancers, including glioblastoma, acute myeloid leukemia (AML), cancers of the female reproductive system (e.g., endometrial, cervical, and ovarian cancers), pancreatic cancer, nasopharyngeal carcinoma, lung cancer, hepatocellular carcinoma, intrahepatic cholangiocarcinoma, testicular germ cell tumors, melanoma, bladder cancer, prostate cancer, breast cancer, renal cancer, gastric cancer, and osteosarcoma [157,158]. The role of m6A in cancer is bidirectional; for instance, in colorectal cancer, SOX2 exerts a pro-cancer effect via METTL3-catalyzed methylation [159]. However, in acute myeloid leukemia (AML), TACC3 exerts pro-cancer effects via ALKBH5-catalyzed demethylation [160].

### 6.1. Quantitative Methods of Detecting m6A Modifications

m6A modifications can be detected by quantitative and localization assays. Quantitative detection techniques include two-dimensional cellulose thin-layer chromatography (2D-TLC) [161], high-performance liquid chromatography combined with tandem mass spectrometry (HPLC/MS-MS) [162] and coupling of liquid chromatography to mass spectrometry (LC-MS) [163].

### 6.2. Localization-Detection Methods for m6A Modifications

Many techniques can be used to detect the localization of m6A [164]. In methylated RNA immunoprecipitation sequencing (MeRIP-seq)/m6A-seq, RNA is randomly interrupted into fragments of 100 nucleotides (nt); then, m6A-containing RNA fragments are captured by m6A-specific antibodies, followed by high-throughput sequencing [165]. To improve the resolution of this method, photo-crosslinking-assisted m6A sequencing (PA-m6A-seq) was developed. This technique can achieve a resolution of 23 nt and effectively improve the accuracy of methylation site assignment [166]. Although PA-m6A-seq technology has improved the accuracy of sequencing, the method still requires the specific peripheral base sequence of m6A to infer the m6A site. In order to accurately determine m6A sites, single-base resolution m6A sequencing methods, m6A individual-nucleotide-resolution crosslinking and immunoprecipitation (miCLIP) [167] and m6A crosslinking and immunoprecipitation (m6A-CLIP) [168], were developed; these techniques use highly specific m6A antibodies. While all of these techniques use m6A antibodies, an antibody-independent method was also developed, m6A-selective allyl chemical labeling and sequencing (m6A-SAC-seq), which allows for single-nucleotide resolution-level analysis of m6A sites on trace RNA samples (~30 ng of poly(A) or rRNA-depleted RNA) [169].

## 7. Conclusions

In this review, we describe the effects of TFs and epigenetic modifications on the regulation of gene expression, including chromatin accessibility, histone modifications, DNA modifications, and RNA modifications. We also describe the relationships of these processes to certain biological processes and diseases and discuss the methods used to detect these processes. At the research level, we hope that these assays will provide scientists with new concepts and hypotheses. At the clinical level, these assays provide a basis for the systematic study of functional significance in biological processes and human diseases. However, further development is needed to develop affordable and highly accurate assays; the ultimate goal is to make these assays clinically applicable.

## Figures and Tables

**Figure 1 biomolecules-13-00304-f001:**
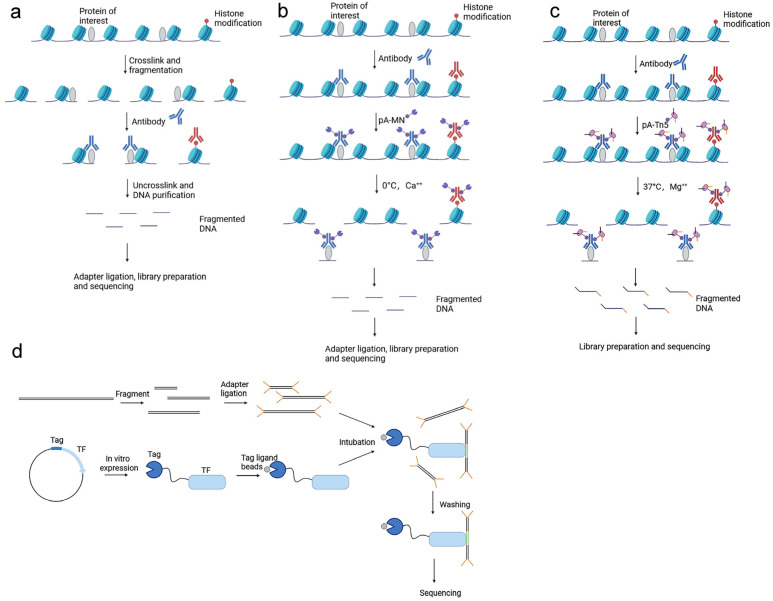
Schematic diagram of ChIP-seq, CUT&RUN, CUT&Tag, and DAP-seq. (**a**) ChIP-seq: formaldehyde crosslinks and fragments the protein and DNA, followed by immunoprecipitation of the protein–DNA complex using antibodies to the target protein, and finally uncrosslinking and purification of the DNA fragment and sequencing of the DNA fragment; (**b**) CUT&RUN: cells are incubated sequentially with antibody and pA-MN, and at 0 °C, Ca++ is added, MNase cleaves DNA on both sides of the binding site, and the fragmented DNA is adapter-ligated, library-prepared, and sequenced; (**c**) CUT&Tag: cells are incubated sequentially with antibody and pA-Tn5, and at 37 °C, Mg++ is added, Tn5 ligates adapters while cleaving DNA on both sides of the target protein, and the fragmented DNA is library-prepared and sequenced; (**d**) DAP-seq: fragmented DNA is ligated to sequencing adapters, then TF fused to Tag is expressed in vitro and bound to ligand-coupled beads, then DNA and transcription factor fusion proteins are incubated, unbound DNA fragments are washed away, and finally transcription factor-bound DNA fragments are enriched for sequencing.

**Figure 2 biomolecules-13-00304-f002:**
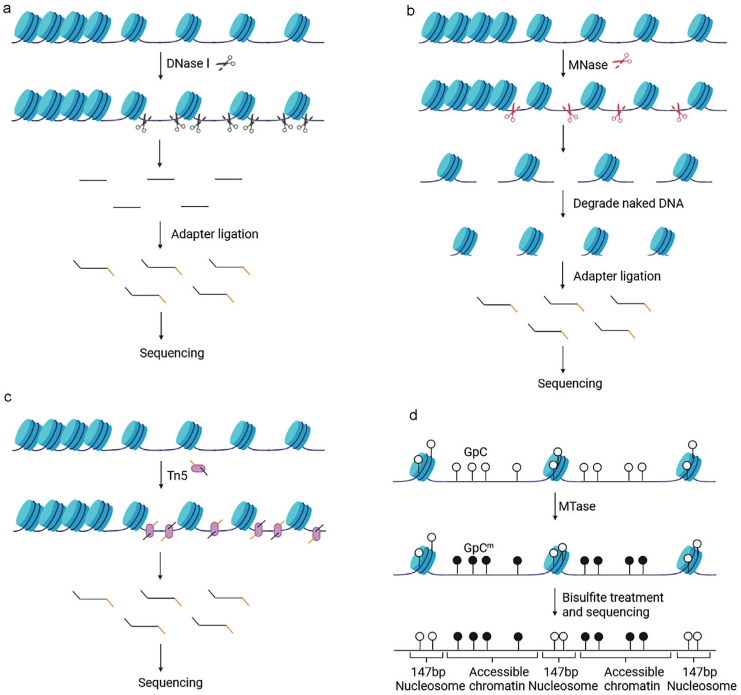
Schematic diagram of DNase-seq, MNase-seq, ATAC-seq, and Nome-seq. (**a**) DNase-seq cleaves accessible chromatin regions characterized by DHSs with DNase I; (**b**) MNase-seq uses MNase to cleave inter-nucleosomal DNA and to degrade the naked accessible DNA; (**c**) ATAC-seq uses Tn5 transposase to ligate sequencing adaptors while cutting DNA fragment; (**d**) Nome-seq uses M.CviPI GpC methyltransferase (MTase) to methylate GpC in accessible chromatin to GpCm, and bisulfite treatment can distinguish GpC from GpCm.

**Figure 3 biomolecules-13-00304-f003:**
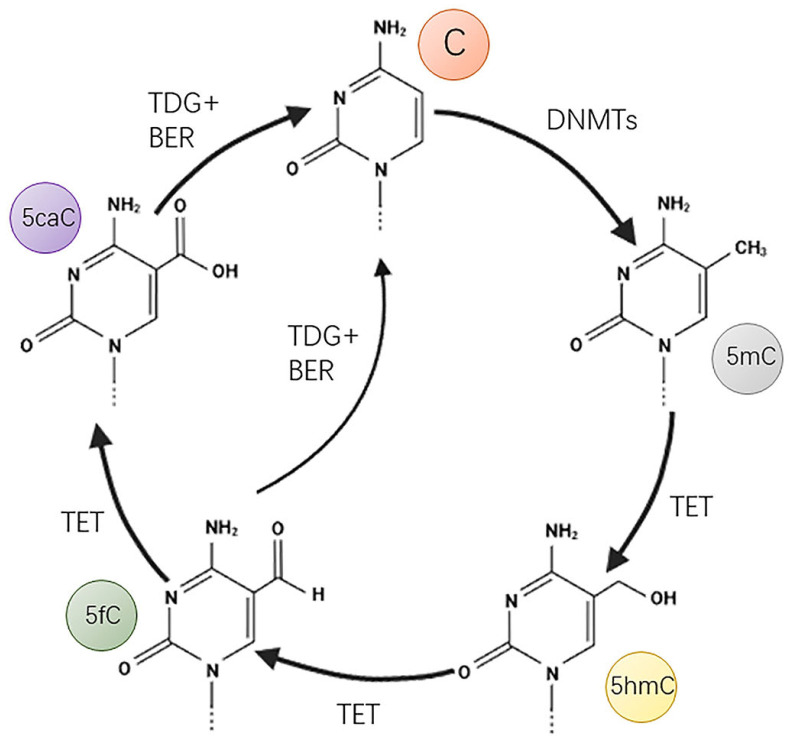
Mechanisms of DNA methylation and active demethylation. DNA methyltransferases (DNMTs) convert C to 5mC, which is progressively oxidized by TET to 5hmC, 5fC, and 5caC. 5fC and 5caC generate unmodified C through TDG- and BER-mediated excision and repair.

**Figure 4 biomolecules-13-00304-f004:**
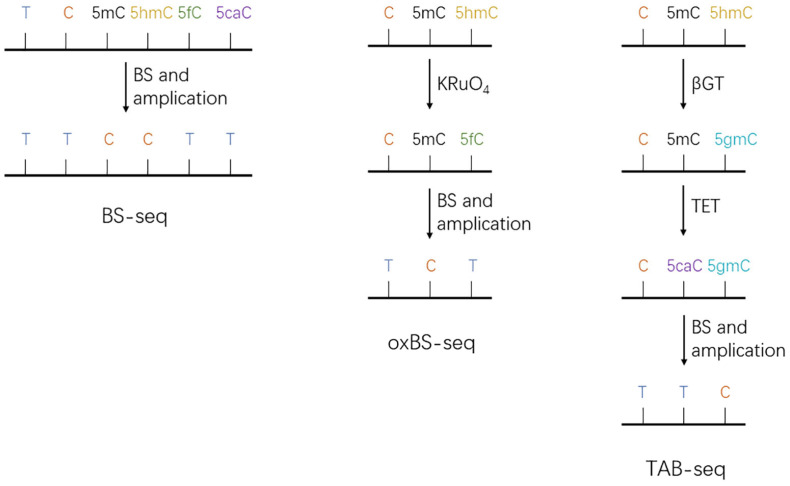
Schematic diagram of BS-seq, oxBS-seq, and TAB-seq. In BS-seq, C/5fC/5caC was read as T and 5mC/5hmC was read as C after sulfite treatment and sequencing. In oxBS-seq, 5hmC was oxidized by KRuO4 to 5fC, which was read as T after bisulfite and sequencing, while 5mC was read as C. In TAB-Seq, β-GT catalyzes the entry of glucose into 5hmC to produce 5gmC, which is sequenced as C, while 5mC is oxidized by excess TET to 5caC, which is sequenced as T.

**Figure 5 biomolecules-13-00304-f005:**
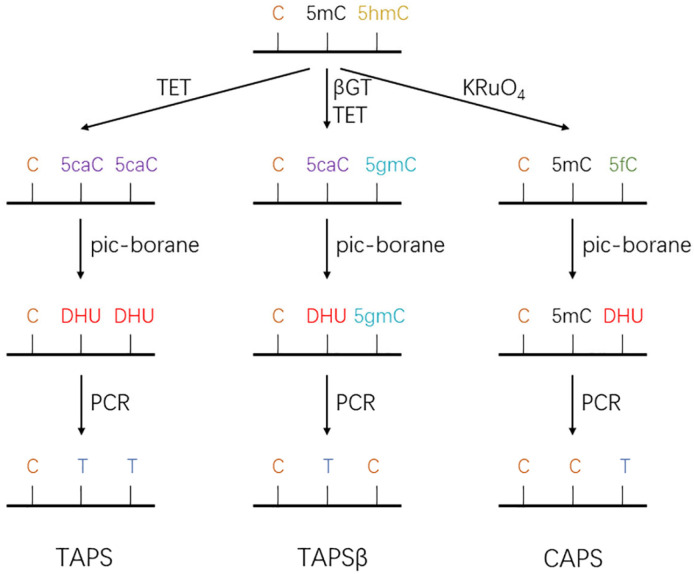
Schematic diagram of TET-assisted pyridine borane sequencing and its derivatives. In TAPS, the TET enzyme oxidizes 5mC and 5hmC to 5caC, and then 5caC is reduced by pyridine borane to DHU. DHU is read as T in subsequent sequencing. In TAPSβ, 5mC is oxidized by TET to 5caC and then reduced by pyridine borane to DHU, and finally read as T, while 5hmC is glycosylated to 5gmC, which is not oxidized by TET and reduced by pyridine borane, and finally 5hmC is read as C. In oxBS-seq, KRuO4 specifically oxidizes 5hmC to 5fC, which is then reduced to DHU by pyridine borane and read as T during sequencing, while 5mC remains unchanged and is read as C.

**Figure 6 biomolecules-13-00304-f006:**
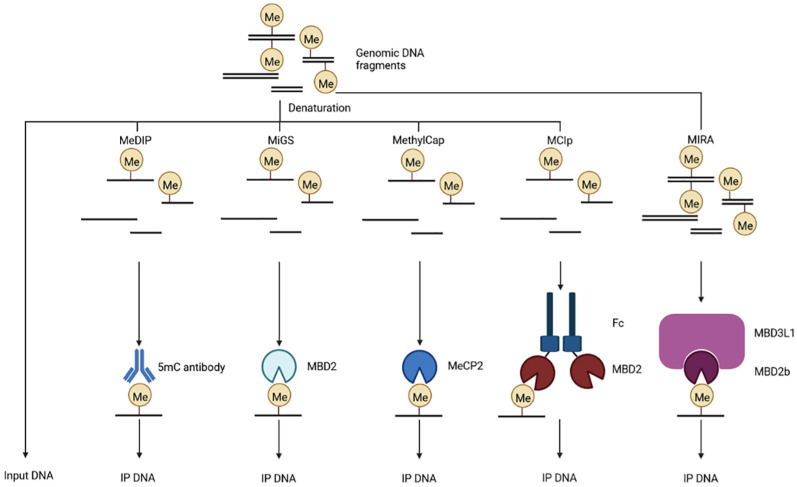
Schematic diagram of methylated DNA immunoprecipitation. MeDIP, MiGS, MethylCap, MCIp, and MIRA use 5mC antibody, recombinant MBD of MBD2 protein, MBD of MeCP2 protein, recombinant antibody-like fusion protein, and MBD2b/MBD3L1 protein complex to precipitate methylated DNA fragments, respectively.

## Data Availability

Not applicable.

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
