# Peer review of "Factors and Methods for the Detection of Gene Expression Regulation"

_biomolecules, 2023, doi:10.3390/biom13020304_

Round 1

Reviewer 1 Report

Overall this is a comprehensive review that is generally well-written and makes a solid contribution   to the field and will be particularly useful to those new to this area of research.

I have a few comments and there are some minor corrections.

Comments

  1. The title could be more accurate.  Before reading the review I thought it would cover methods that look directly at gene expression and the directed mutation of the cis-acting elements that quantitatively regulate the levels gene expression.  More emphasis should be given to the fact that it discusses the factors that influence gene expression rather than the process of gene expression itself.

  1. When discussing histone modifications (section 4, p7) it might be more systematic to refer to the writers, readers and erasers that are involved with the various marks.  This is done in the later section on the analysis of RNA modifications. 

3. It is my impression that third generation sequencing platforms will be increasingly used to directly determine DNA modifications.  This section is surprisingly concise compared to the extensive sections on the methods involving chemical modification followed by sequencing.  A bit more on the accuracy of these third gen methods, such as why the kinetics of incorporation varies  when there is a modified base in the template and how reliable this is, would help when making decisions about the methods to use.

Minor points

  1. Section 2.1.3 DNA affinity purification sequencing:  needs more detail about the Tag and how it works
  2. P7 line 260-261:  HDAC is a histone deacetylase not an acetylase.
  3. P8 line 294: histone rather than histidine.
  4. P8 line 313/4;  need to explain more clearly what is meant by long range histone modification combinations.
  5. P9. Line 350:  What sort of alterations in DNA methylation have been detected. Increased levels or altered patterns/profiles?
  6. P10 line 377. Oxidised rather than oxidated?
  7. P14 line 536/537. In what way were the 3’UTRs elongated?
  8. P15 line 588.  Test rather than text?

Author Response

We greatly appreciate the valuable comments. We have revised the manuscript according to the suggestions and re-submitted a new version of the manuscript. Your concerns have been addressed in the revised manuscript as described below:

Comment 1: The title could be more accurate.  Before reading the review I thought it would cover methods that look directly at gene expression and the directed mutation of the cis-acting elements that quantitatively regulate the levels gene expression.  More emphasis should be given to the fact that it discusses the factors that influence gene expression rather than the process of gene expression itself.

Response 1: These factors are involved in gene expression regulation and the mechanisms by which they are involved in gene expression regulation are described in thereview, with emphasis on their detection methods. It is emphasized that they are involved in gene expression regulation and affect gene expression.

Comment 2: When discussing histone modifications (section 4, p7) it might be more systematic to refer to the writers, readers and erasers that are involved with the various marks.  This is done in the later section on the analysis of RNA modifications. 

Response 2: We have demonstrated histone acetyltransferase and histone deacetylase in P7 line 268-269 and P7 line 269-272.We have demonstrated histone methyltransferase and histone demethylase in P7 line 279-281 and P7 line 284-285.We have demonstrated protein kinase and phosphatase in P7 line 294-295 and P7 line 295-296.

Comment 3: It is my impression that third generation sequencing platforms will be increasingly used to directly determine DNA modifications.  This section is surprisingly concise compared to the extensive sections on the methods involving chemical modification followed by sequencing.  A bit more on the accuracy of these third gen methods, such as why the kinetics of incorporation varies  when there is a modified base in the template and how reliable this is, would help when making decisions about the methods to use.

Response 3:  We have  added descriptions about nanopore sequencing and SMRT sequencing in P13 line 517-522 and P14 line 537-540.

Point 1: Section 2.1.3 DNA affinity purification sequencing:  needs more detail about the Tag and how it works

Response 1: We have  added descriptions about  DNA affinity purification sequencing in P3 line 120-125.                                    

Point 2: P7 line 260-261:  HDAC is a histone deacetylase not an acetylase.

Response 2: We 've corrected the typo.

Point 3: P8 line 294: histone rather than histidine.

Response 3: We 've corrected the typo.

Point 4: P8 line 313/4;  need to explain more clearly what is meant by long range histone modification combinations.

Response 4:  We have  added descriptions about  long range histone modification combinations in P8 line 323-326.                                       

Point 5: P9. Line 350:  What sort of alterations in DNA methylation have been detected. Increased levels or altered patterns/profiles?

Response 5:  In the process of human aging,DNA methylation  either increasing (hypermethylation) or decreasing (hypomethylation) at the CpG site. Meanwhile, altered DNA methylation patterns are associated with a broad range of age-related diseases.

Point 6: P10 line 377. Oxidised rather than oxidated?

Response 6: The "be oxidated to" has the same meaning as "be oxidized to".

Point 7: P14 line 536/537. In what way were the 3’UTRs elongated?

Response 7:  3′UTR m6A modification correlates with alternative
polyadenylation (APA) and methylated transcripts tend to be coupled with the proximal APA site and thus have shortened 3′UTRs.The knockdown of VIRMA and METTL3 makes the level of m6A decrease and thus elongated 3′UTR.

Point 8: P15 line 588.  Test rather than text?

Response 8: We have made changes in this review in P15 line 613-614.     

Reviewer 2 Report

Here, the authors present a literature review of mechanisms and methods for the detection of gene expression regulation. This topic is of interest to a wide range of researchers and must be written in the depth. However, the present form of the review is very much superficial, and just describes the name of the methods and where it has been used.  Although the title says the mechanisms and methods for gene regulation, this review contains only the names of the methods and for which purposes these are used. There is no description of the mechanisms of gene regulation in this review, the word "mechanisms"  does not fit in the title of this review. There are already various articles covering these methods in detail.  Authors should emphasize writing methods in detail. Include tables comparing these methods, and the limitations of each method, which would be beneficial for early carrier researchers.

Author Response

Comment: Here, the authors present a literature review of mechanisms and methods for the detection of gene expression regulation. This topic is of interest to a wide range of researchers and must be written in the depth. However, the present form of the review is very much superficial, and just describes the name of the methods and where it has been used.  Although the title says the mechanisms and methods for gene regulation, this review contains only the names of the methods and for which purposes these are used. There is no description of the mechanisms of gene regulation in this review, the word "mechanisms"  does not fit in the title of this review. There are already various articles covering these methods in detail.  Authors should emphasize writing methods in detail. Include tables comparing these methods, and the limitations of each method, which would be beneficial for early carrier researchers.

Response:                                                                                                              

Firstly, we sincerely appreciate the valuable comments. We have introduced at the beginning of each section how this factor is involved in the regulation of gene expression. The mechanism part is located in front of the method.               

Secondly,  for the description of each methodological approach, I have first described its principles and illustrated them with diagrams, and then have described the advantages and disadvantages as well as its application in clinical practice, if any.    

You can find these in the revised manuscript.   

Reviewer 3 Report

I read with great interest the manuscript by Wang & Li et al. entitled “Mechanisms and methods for the detection of gene expression regulation”. In this manuscript the authors summarized the key methods for detecting the biological roles of transcription factors and epigenetics status for regulating gene expressions. 

For the broad interest of readers and tracking of latest research topics, by addressing the following comments, this reviewer will fully support the publication of this review on Biomolecules

1.     English writing should be improved.

2.     “In 2007, Johnson et al.[7] used ChIP-seq to discover genome-wide bind sites for the transcription factor neuron-restrictive silencer factor(NRSF) to DNA; this represented the first application of ChIP-seq to the analysis of TFs. “I think the first application of TF ChIP-seq is presented in the study of High-Resolution Profiling of Histone Methylations in the Human Genome for CTCF on May 2007, while the reference mentioned by the author was published on Jun 2007. 

3.     This reviewer is not sure if reference 29 is related to the sentence of “dual-luciferase reporter assays are then used to verify 135 the transcriptional activity of the target TF[29,30].”

4.     This reviewer is not sure if reference 39 is a fundamental study for the statement of “Subsequent research found that functional cis-acting elements, such as enhancers, 167 promoters, silencers and insulators, are coupled to DNase I hypersensitive sites [37–39].”

5.     This reviewer cannot agree the data analysis is still immature for ATAC-seq in 2023 as stated by the author “(1) immature sequence data analysis [42]”, by which the reference 42 was published 2014 and ATAC-seq was published on 2013. Also regarding the limitation of  “(3) sequencing can be contaminated by mitochondrial DNA[54]”, the study of The landscape of accessible chromatin in mammalian preimplantation embryos already proposed a solution. 

6.     I think the paragraph 4.4 is duplicated for 4.5. 

7.     “ChIP-seq, CUT&RUN, and CUT&Tag are based on histone modifications of interest to determine their binding sites in the genome. “Classical references should be added. 

8.     Generally, I feel the part of RNA modifications are not highly related to the above parts of TFs, histone modifications, and DNA-methylations, which are key topics of epigenetics. In contrast, RNA modifications seem to be in another territory. For the tracking of epigenetics and gene regulation research topics, I highly suggested the authors replace the RNA modifications part for the 3D genome methods, such as high-resolution microscope-based images and genome-wide sequencing-based methods such as Hi-C (single-cell Hi-C) or TF/histone marker center views of ChIA-PET, HiChIP, HiTrAC, ChIATAC, HiCAR, etc.

Author Response

We greatly appreciate the valuable comments. We have revised the manuscript according to the suggestions and re-submitted a new version of the manuscript. Your concerns have been addressed in the revised manuscript as described below:

Comment 1. English writing should be improved.

Response 1:We have corrected the grammatical, styling, and typos found in our manuscript.

Comment 2. “In 2007, Johnson et al.[7] used ChIP-seq to discover genome-wide bind sites for the transcription factor neuron-restrictive silencer factor(NRSF)to DNA; this represented the first application of ChIP-seq to the analysis of TFs. “I think the first application of TF ChIP-seq is presented in the study of High-Resolution Profiling of Histone Methylations in the Human Genome for CTCF on May 2007, while the reference mentioned by the author was published on Jun 

Response 2:We agree with you and we have deleted "this represented the first application of ChIP-seq to the analysis of TFs".

Comment 3: This reviewer is not sure if reference 29 is related to the sentence of “dual-luciferase reporter assays are then used to verify 135 the transcriptional activity of the target TF[29,30].”

Response 3:In reference 29, dualluciferase reporter assays were performed to validate the regulatory relationships between transcription factor  EGR2 and potential target genes IGF2BPs.

Comment 4: This reviewer is not sure if reference 39 is a fundamental study for the statement of “Subsequent research found that functional cis-acting elements, such as enhancers, 167 promoters, silencers and insulators, are coupled to DNase I hypersensitive sites [37–39].”

Response 4:We re-read reference 39 and it is a sentence in the literature that is relevant to “Subsequent research found that functional cis-acting elements, such as enhancers, 167 promoters, silencers and insulators, are coupled to DNase I hypersensitive sites [37–39].”. But this piece of literature should not be used as a citation and we have made changes in P5 line 174-176.

Comment 5: This reviewer cannot agree the data analysis is still immature for ATAC-seq in 2023 as stated by the author “(1) immature sequence data analysis [42]”, by which the reference 42 was published 2014 and ATAC-seq was published on 2013. Also regarding the limitation of  “(3) sequencing can be contaminated by mitochondrial DNA[54]”, the study of The landscape of accessible chromatin in mammalian preimplantation embryos already proposed a solution. 

Response 5:We agree with you and we have made changes in P6 line 222-224.

Comment6: I think the paragraph 4.4 is duplicated for 4.5. 

Response 6:4.4 is based on histone modifications of interest to determine their binding sites in the genome,while 4.5 Mass spectrometry (MS)-based proteomics is a tool for identifying histone modifications species,and it can also  discover new histone modifications.

Comment 7: “ChIP-seq, CUT&RUN, and CUT&Tag are based on histone modifications of interest to determine their binding sites in the genome. “Classical references should be added. 

Response 7:We have added in P8 line 314-315.

Comment 8: Generally, I feel the part of RNA modifications are not highly related to the above parts of TFs, histone modifications, and DNA-methylations, which are key topics of epigenetics. In contrast, RNA modifications seem to be in another territory. For the tracking of epigenetics and gene regulation research topics, I highly suggested the authors replace the RNA modifications part for the 3D genome methods, such as high-resolution microscope-based images and genome-wide sequencing-based methods such as Hi-C (single-cell Hi-C) or TF/histone marker center views of ChIA-PET, HiChIP, HiTrAC, ChIATAC, HiCAR, etc.

Response 8:RNA modification belongs to epigenetics and is involved in the regulation of gene expression in several ways, so it is still necessary for RNA modification to appear in this review. As for 3D genome methods, I can add them if you think it is necessary.

Round 2

Reviewer 2 Report

As I asked in my previous comment, the present form of the review is very superficial and describes the name of the methods and where it has been used. Although the title says the mechanisms and methods for gene regulation, this review contains only the names of the methods and for which purposes these are used. There is no description of the mechanisms of gene regulation in this review, the word "mechanisms" does not fit in the title of this review. I would recommend using the word "factors" instead of the word "mechanisms".  This review would be beneficial for early carrier scientists. 

Author Response

Responses to  Reviewer comments

Manuscript ID: 2174200

Title: Mechanisms and methods for the detection of gene expression regulation 

Comments:

As I asked in my previous comment, the present form of the review is very superficial and describes the name of the methods and where it has been used. Although the title says the mechanisms and methods for gene regulation, this review contains only the names of the methods and for which purposes these are used. There is no description of the mechanisms of gene regulation in this review, the word "mechanisms" does not fit in the title of this review. I would recommend using the word "factors" instead of the word "mechanisms".  This review would be beneficial for early carrier scientists.

Response:

We greatly appreciate  the  valuable comments. We have revised the manuscript according to the reviewer’s suggestions and re-submitted a new version of the manuscript. We have changed the "mechanisms" to the word "factors" in P1 line 2,P1 line 14 and  P1 line39.